# Fertility of Pedicellate Spikelets in Sorghum Is Controlled by a Jasmonic Acid Regulatory Module

**DOI:** 10.3390/ijms20194951

**Published:** 2019-10-08

**Authors:** Nicholas Gladman, Yinping Jiao, Young Koung Lee, Lifang Zhang, Ratan Chopra, Michael Regulski, Gloria Burow, Chad Hayes, Shawn A. Christensen, Lavanya Dampanaboina, Junping Chen, John Burke, Doreen Ware, Zhanguo Xin

**Affiliations:** 1Plant Stress and Germplasm Development Unit, Cropping Systems Research Laboratory, U.S. Department of Agriculture-Agricultural Research Service, Lubbock, TX 79415, USA; gladman@cshl.edu (N.G.); yjiao@cshl.edu (Y.J.); rchopra@umn.edu (R.C.); gloria.burow@ars.usda.gov (G.B.); chad.hayes@ars.usda.gov (C.H.); lavanya.dampanaboina@ttu.edu (L.D.); junping.chen@ars.usda.gov (J.C.); john.burke@ars.usda.gov (J.B.); 2Cold Spring Harbor Laboratory, Cold Spring Harbor, NY 11724, USA; leeyk@nfri.re.kr (Y.K.L.); zhangl@cshl.edu (L.Z.); regulski@cshl.edu (M.R.); 3Plasma Technology Research Center, National Fusion Research Institute, 37, Dongjangsan-ro, Gunsan-si, Jeollabuk-do 54004, Korea; 4Current address: Department of Agronomy and Plant Genetics, University of Minnesota, St. Paul, MN 55108, USA; 5Chemistry Research Unit, USDA-ARS, 1700 S.W. 23RD DRIVE, Gainesville, FL 32608, USA; shawn.christensen@ars.usda.gov; 6U.S. Department of Agriculture-Agricultural Research Service, NEA Robert W. Holley Center for Agriculture and Health, Cornell University, Ithaca, NY 14853, USA

**Keywords:** transcriptional regulators, plant development, jasmonic acid signaling, gene expression

## Abstract

As in other cereal crops, the panicles of sorghum (*Sorghum bicolor* (L.) Moench) comprise two types of floral spikelets (grass flowers). Only sessile spikelets (SSs) are capable of producing viable grains, whereas pedicellate spikelets (PSs) cease development after initiation and eventually abort. Consequently, grain number per panicle (GNP) is lower than the total number of flowers produced per panicle. The mechanism underlying this differential fertility is not well understood. To investigate this issue, we isolated a series of ethyl methane sulfonate (EMS)-induced *multiseeded* (*msd*) mutants that result in full spikelet fertility, effectively doubling GNP. Previously, we showed that MSD1 is a TCP (Teosinte branched/Cycloidea/PCF) transcription factor that regulates jasmonic acid (JA) biosynthesis, and ultimately floral sex organ development. Here, we show that *MSD2* encodes a lipoxygenase (LOX) that catalyzes the first committed step of JA biosynthesis. Further, we demonstrate that MSD1 binds to the promoters of *MSD2* and other JA pathway genes. Together, these results show that a JA-induced module regulates sorghum panicle development and spikelet fertility. The findings advance our understanding of inflorescence development and could lead to new strategies for increasing GNP and grain yield in sorghum and other cereal crops.

## 1. Significance

Through a single base pair mutation, grain number can be increased by ~200% in the globally important crop *Sorghum bicolor*. This mutation affects the expression of an enzyme, MSD2, that catalyzes the jasmonic acid pathway in developing floral meristems. The global gene expression profile in this enzymatic mutant is similar to that of a transcription factor mutant, *msd1*, indicating that disturbing any component of this regulatory module disrupts a positive feedback loop that occurs normally due to regular developmental perception of jasmonic acid. Additionally, the MSD1 transcription factor is able to regulate *MSD2* in addition to other jasmonic acid pathway genes, suggesting that it is a primary transcriptional regulator of this hormone signaling pathway in floral meristems.

## 2. Introduction

Sorghum (*Sorghum bicolor* (L.) Moench) is a crop plant domesticated in northern Africa ~6000 years ago [1,2]. A C_4_ grass with robust tolerance to drought, heat, and high-salt conditions, sorghum is the fifth most agriculturally important crop in terms of global dedicated acreage and production quantity. It also serves as a useful model for crop research due to its completely sequenced compact genome (~730 Mb) [3] and similarity to the functional genomics capabilities of maize, sugarcane, and other bioenergy grasses comprising more convoluted genomes.

Increasing grain yield has always been a high priority for breeders. Increasing grains per panicle (GNP) and optimizing panicle architecture represent feasible goals for modern gene editing in sorghum [4,5]. GNP and seed head architecture are related traits, with origins in early stages of inflorescence development [6,7]. Sorghum forms a determinant panicle that manifests at the end of the shoot meristem, with nodes regularly extending throughout from which secondary and tertiary branches emanate [8]. A terminal trio of spikelet florets are attached through a pedicel to these branches: one sessile spikelet (SS) that is fertile and two pedicellate spikelets (PSs) that fail to generate mature pistils and sometimes anthers, which results in an inability of PSs to fertilize. They will ultimately senesce during grain filling instead of becoming viable seed [6]. Below this terminal spikelet group, several pairs of SSs and PSs populate the branches down to the nodes.

Jasmonic acid (JA) is a plant hormone derived from α-linolenic acid and shares structural similarities to animal prostaglandins [9,10]. JA plays roles in organ development, as well as biotic and abiotic response signaling mechanisms [7,11,12,13,14], spikelet formation in rice [15], and sex determination in maize [13,16,17]. In sorghum, the TCP family transcription factor MSD1 (*multiseeded 1*) controls PS fertility [7]. MSD1 is expressed in a narrow spatiotemporal window within the developing panicle in wild type (WT), BTx623; its expression is dramatically reduced in ethyl methane sulfonate (EMS)-induced *msd1* mutants. Many genes involved in JA biosynthesis, including 12-oxophytodeinoate reductase 3 (*OPR3*) [18,19], allene oxide synthase (*AOS*) [20], cytochrome P450 [21,22], and lipoxygenase (*LOX*) [23], are also downregulated in *msd1* mutants. MSD1 is thought to activate the programmed cell death pathway through activation of JA biosynthesis, which destines the PS to abortion.

In this study, we characterized another *msd* mutant, *msd2*, from the same publicly available sorghum EMS population in which *msd1* was identified [24]. *MSD2* encodes a 13-lipoxygenase that catalyzes the conversion of free α-linolenic acid (18:3) to 13(S)-hydroperoxylinolenic acid (13-Hpot), the first committed step of the JA biosynthetic pathway [11,25]. As with *msd1*, mutants in *msd2* exhibit complete spikelet fertility for both SSs and PSs, resulting in seed formation from both flower types. Multiple independent alleles were discovered for *msd2*, including nonsense and missense mutations within the LOX functional domain. Mutants in *msd1* and *msd2* exhibit similar regulatory network profiles, including downregulation of JA pathway genes and other expression cascades related to developmental and cellular restructuring. Finally, MSD1 is capable of activating *MSD2* expression, as well as regulating other gene networks in trans, leading to the *multiseeded* phenotype. Taken together, our findings demonstrate that MSD2, along with MSD1, modulates the JA pathway during sorghum sex organ development.

## 3. Results

### 3.1. MSD2 Encodes a Lipoxygenase in the Jasmonic Acid Biosynthetic Pathway

*Sorghum bicolor* (L.) Moench plants manifesting the *multiseeded* phenotype were identified from a collection of EMS-induced single nucleotide polymorphisms (SNP) [24]. *MSD1*, which encodes a TCP (*Teosinte branched 1* (*TB1*), *Cycloidea* (*Cyc*), and *Proliferating Cell nuclear antigen binding Factor* (*PCF*)) [26,27,28] transcription factor, was the first to be characterized, revealing a role in controlling bioactive JA levels in developing floral meristems [7]. To identify additional causative alleles, we subjected seventeen different *msd* mutants to whole-genome sequencing followed by comparative variant calling analysis. Three of these independent alleles, *msd2-1, -2, and -3*, localized to SORBI_3006G095600 (Sb06g018040) [7], which encodes a class II 13-lipoxygenase that shares >95% amino acid identity with the maize *tasselseed 1* (*TS1*) gene [16]. SORBI_3006G095600 is syntenic to *TS1* and is the closest related maize orthologue based on a maximum likelihood phylogenetic analysis (Appendix A). The *msd2-1* mutant harbors a nonsense mutation (peptide residue Q402*) and *msd2-2* a missense mutation (peptide residue A423V), respectively, within the lipoxygenase (LOX) domain (Figure 1A). The *msd2-3* allele contains the same mutation as *msd2-1*, but the lines are not siblings, as evidenced by the lack of other shared SNPs.

Lipoxygenases catalyze linolenic acid to hydroperoxyoctadecadienoic acid in the initial committed step of the JA biosynthetic pathway [11]. There are 12 LOX paralogs in *Sorghum bicolor* (Figure 1B), which exhibit different patterns of tissue-specific expression in WT BTx623 plants. *MSD2* is expressed at lower levels overall than other LOX genes [29,30,31,32], but is more strongly expressed in developing panicles than its 13-LOX paralogs SORBI_3007g210400 and SORBI_3001G483400 (Figure 1C); only SORBI_3004G078600 is more strongly expressed at particular stages. Thus, like *MSD1*, *MSD2* operates under low levels of expression in specific tissues within developing meristems. EMS-induced SNP mutant lines exist for the other 13-LOX paralogs (Appendix A), including a nonsense mutation in the more highly expressed SORBI_3004G078600, but no *multiseeded* phenotype has been observed in any of these lines. This suggests that MSD2 is a specific and necessary LOX isoform involved in the JA pathway, which controls floral organ progression.

*MSD2* mutants display the same floral phenotype as *msd1*: complete development of anthers and ovaries in both PSs and SSs. Electron micrographs of developing floral spikelets revealed that the developmental pattern of *msd2* is similar to that of WT [7] (Figure 1D), but like *msd1*, the end result is complete floral fertility of all spikelets with near 100% grain filling, increasing the GNP of the mutant (Figure 1E), although the *msd2* seeds from both PS and SS are smaller than those of WT. Dissected images of PSs also show that *msd2* has the same full pistil development phenotype as *msd1* in contrast to WT PSs that lack mature gynoecia (Appendix A). The only other consistent agronomic difference between *msd2* and WT plants were a slight increase in initial root growth rate in the mutant (Appendix A).

### 3.2. The msd2 Phenotype is Rescued by Exogenous Methyl-JA Treatment

To determine whether exogenous application of JA could rescue the *msd2* phenotype, as it does in *msd1* mutants, we pipetted 1 mM methyl-JA (Me-JA) directly down the whorls of young WT and *msd2* mutant plants. This chemical treatment restored PS infertility (Figure 2). Panicle size was reduced in Me-JA treated plants, as was branch length and number. Panicle emergence was also delayed in all genotypes relative to untreated or negative control plants, likely due to other developmental signaling effects and inhibition of cell expansion caused by the introduction of exogenous jasmonates [33,34].

### 3.3. MSD2 Regulatory Networks Are Similar to MSD1

Transcriptomic data indicated that many JA biosynthetic pathway genes, including all *LOX* paralogs, were coordinately downregulated in stage 4 SS and PS tissues of developing *msd2* panicles (Figure 3A,B). Within these tissues, the global transcriptomic profile of genes downregulated in *msd2* revealed Gene Ontology (GO) term enrichment for proteins involved in oxylipin biosynthesis, as well as reorganization of cellular structure (Appendix A), including members of the glycoside hydrolase, lipid transferase, and cellulose synthase families. Genes upregulated within stage 4 PS and SS tissues of *msd2* were enriched for functions related to system development, an ontological group consisting of transcription factors involved in developmental signaling and progression (Appendix A).

Comparison of *msd1* and *msd2* transcriptomes revealed conserved GO enrichment categories, with little difference in expression of JA biosynthetic and signaling genes between mutants in the TCP transcription factor and lipoxygenase components of the hormone pathway. Principal component analysis (PCA) of JA pathway gene expression in both *msd1* and *msd2* showed that the greatest variance involved particular JA biosynthesis genes, predominantly cytochrome, jasmonate methyl transferases, *OPC-8*, *OPR*, and *LOX* genes (Figure 3C). Early-stage meristems (stage 1 and stage 3) exhibited the least variance between the *msd1* and *msd2* transcriptional profiles, whereas stage 4 and 5 inflorescences and spikelets made the greatest contribution to PCA dimensionality. PCA eigenvectors also indicated that the transcriptional divergence between WT and *multiseeded* plants occurs around stage 4 and continues through maturation in stage 5 (Figure 3D). A set of 149 genes identified by Jiao et al. (2018) as putative regulatory targets of MSD1 was strongly downregulated in *msd2* in either stage 4 or stage 5 tissues (Appendix A). Again, dimensional analysis of RNA-seq data revealed little variation between *msd1* and *msd2*, and indicated that stage 4 meristem marks the moment of demarcation between *multiseeded* and WT plants (Appendix A).

Motif analysis of JA biosynthetic and signaling genes revealed enrichment for various developmental and environmentally responsive DNA-binding domains, specifically the AP2, BZR (brassinazole-resistant family), bZIP, and WRKY families, as well as TCP proteins (Appendix A). A similar analysis of the 149 candidate MSD1 regulatory targets yielded a significant enrichment for CG-rich motifs strongly recognized by TCP, AP2, MYB, and E2F (specifically Della) transcription factors (Appendix A).

### 3.4. MSD2 Is Regulated by the TCP Transcription Factor MSD1

To evaluate if MSD1 directly regulates components of the JA biosynthetic pathway, Yeast 1-hybrid (Y1H) analysis was performed to determine whether MSD1 directly regulates *MSD2* in trans. Indeed, MSD1 bound to the sequence upstream of the *MSD2* transcriptional start site (TSS). MSD1 also bound sequence upstream of its own TSS (Figure 4A). These observations are consistent with the idea that MSD1 controls expression of both itself and *MSD2*.

In a less biased investigation of MSD1 regulatory targets, we conducted DNA Affinity Purification sequencing (DAP-seq) [35] analysis using Illumina short-read libraries constructed from developing floral meristem tissues and incubated with bacterially-expressed GST-tagged MSD1 proteins. The full list of 2730 identified peaks with their nearest annotated genes is provided as Appendix A. Motif analysis of peaks localized near TSSs were only enriched for the canonical TCP DNA binding motif (GTGGGNCC) bound by other plant TCP proteins (Figure 4B) [35,36,37]. Comparing these peaks with RNA-seq data revealed that 124 of the genes associated with these peaks were at least two-fold downregulated in stage 4 PS or SS tissues of *msd1* mutants, and the size of this candidate gene list increased when we included genes downregulated in any meristem tissue stage. Based on homology to JA pathway genes from other plant species, a subset of these downregulated genes is involved in JA biosynthesis or signaling (Figure 4C), including the *LOX* genes SORBI_3008G157900 and SORBI_3007G210400. Additionally, several genes associated with DAP-seq peaks were among the 149 MSD1 regulatory targets identified by Jiao et al. (2018): they are SORBI_3001G012200 (cytochrome P450), SORBI_3001G202600 (glutamyl-tRNA reductase), SORBI_3002G227700 (lipase), SORBI_3003G061900 (zinc finger), SORBI_3007G004600 (ferredoxin-type iron-sulfur binding domain containing protein), SORBI_3007G035600 (MSP domain containing protein), SORBI_3009G032600 (peroxidase), and SORBI_3009G100500 (WRKY).

The majority of DAP-seq peaks were localized more than 2000 base pairs upstream or downstream of the nearest annotated gene TSS, suggesting that they represent MSD1-targeted enhancer regions. When we applied motif analysis to all 2730 peaks, we identified additional DNA binding sequences. Several of the most significant motifs are recognized by other environmentally responsive and developmental transcription factors, including AP2, WRKY, HOMEOBOX, bZIP, and MYB (Figure 4D). Transcripts downregulated in *msd1* panicles that were also associated with MSD1 DAP-seq peaks included homologs of developmental signaling gene products, such as *Ramosa3* and *Embryonic Flower 1*, as well as several WRKY, AP2/ERF, and ZINC-finger transcription factors.

## 4. Discussion

*MSD2* functions as an essential developmental gatekeeper in floral sex organ development; *MSD2*-deficient plants exhibit 100% flower fertility and grain filling, culminating in higher GNP. Mutant *msd2* panicles have similar transcriptomic profiles to mutants in the TCP transcription factor *MSD1*, suggesting that their respective phenotypes are both the result of disrupting an enzymatically-controlled feedback loop. MSD1 has the capacity to bind the promoter of *MSD2*, as well as promoters and more distant genetic elements associated with developmental and JA pathway genes, including those encoding other LOX-domain containing proteins.

JA is integral to environmental responsiveness and developmental progression; regulators of JA, JA biosynthetic genes, and JA signaling proteins influence pest/pathogen sensitivity [38,39], wound response [22,40,41], cell expansion [33], and sex determination and floral organ progression (specifically, anther and pistil development) [7,15,16,17,22,27,42,43]. Barley, rice, and maize display complex mechanisms of floral development. These modules have been genetically dissected via mutant analyses of homeobox, AP2/ERF, and MADS-box transcription factors as well as JA signaling genes and biosynthetic lipases. Notably, *MSD2* influences pistil progression in sorghum spikelets and the maize ortholog *TS1* controls pistillate determinacy in tassels. Molecular interpretation of these regulatory network ensembles reveals that repression of spikelet fertility in grasses is the norm and is modulated through one or a number of hormonal pathways in a given Poaceae lineage, which include JA and auxin [15,44,45,46].

Specifically, the role of *MSD2* in regulating floral organ fertility in sorghum is analogous to those of other LOX domain–containing proteins from other plant models [11,16,17,25,47]; multiple paralogs exist and exhibit variable expression through the stages of meristem development, indicating potential redundancy and narrow spatiotemporal expression of key *LOX* genes during development. Despite the higher expression of some LOX paralogs in *msd* meristems, exogenous Me-JA is sufficient to rescue the *multiseeded* phenotype, indicating that *msd2* is specifically responsible for sufficient JA signaling in the meristem cells that differentiate into male and female organs in PS and SS tissues.

Furthermore, the *msd2* RNA-seq data reveals the specific JA module of developmental control within sorghum; the data confirms the previous observation that the lack of a functional lipid enzyme can dismantle a regulatory network, resulting in observable downregulation of other biosynthetic pathway genes [10,48,49]; ablating elements of the JA pathway triggers the disruption of a positive feedback loop that would otherwise progress normally due to regular developmental perception of JA. In the case of *msd2*, this yielded a transcriptomic profile similar to that of the TCP transcription factor mutant *msd1* across developmental time points in immature meristems. Furthermore, protein–protein interaction and GO enrichment analyses of *msd1* and *msd2* gene networks in developing panicles suggest the existence of a distinct molecular avenue that leads to elevated GNP by diverting the expression of cellular restructuring genes and shunting to alternate developmental cascades mediated by other transcription factors and influenced by other hormone pathways.

MSD1 can bind to the *MSD2* promoter and activate gene expression. Consistent with this, expression analysis also revealed reduced levels of *MSD2* transcript in the *msd1* mutants [7]. In addition, DAP-seq analysis showed that MSD1 associates with other JA biosynthetic and signaling genes, including both 9- and 13-LOX paralogs of *MSD2*. Additionally, we identified potential enhancer binding regions for MSD1 that also exhibit enrichment for motifs bound by other developmental and environmental response transcription factors, suggesting that MSD1 participates in a mixed model of enhancer organization throughout the genome [50]. However, further chromatin conservation and architecture analyses will be required to elucidate the complete enhancer profile of this and other TCP proteins. It should be noted that although the sorghum *Ramosa3* ortholog was downregulated, several other trehalose phosphatase genes, in addition to *Clavata3*, were strongly upregulated in the mutants, suggesting that in *msd* mutants there is a diverting of the developmental signaling networks that canonically dictate sex organ determinacy in some plant lineages. The MSD1 DNA-binding data, together with the transcriptomic overlap of *msd1* and *msd2* mutants, provide further support of a model in which JA is responsible for regulating floral sex organ fate in *Sorghum bicolor*, and MSD1 is a major regulator of gene expression in this developmental schema (Appendix A).

## 5. Materials and Methods

### 5.1. Identification of MSD2 and Phenotypic Evaluation

Seventeen *msd* mutants were isolated from an ethyl methane sulfonate (EMS) population [51] and grown in the field of the USDA-ARS Cropping Systems Research Laboratory at Lubbock, TX (33′39” N, 101′49” W). High-quality DNA was extracted [52] from confirmed *msd* lines and submitted for whole-genome sequencing at Beijing Genomic Institute (https://www.bgi.com/us/). Reads were trimmed and aligned to the sorghum reference genome v1.4 with Bowtie2 [53]. SNP calling was carried out on reads with PHRED >20 using SAMtools [54] and BCFtools; read depth was set from 3 to 50. Only homozygous G/C to A/T SNP transitions were filtered through to prediction by the Ensembl variation predictor [55]. Functional annotations of genes along with homology and syntenic analyses were derived from the Gramene database release 39 [56]. Phylogenetic analysis (boxshade and trees) was performed using MUSCLE alignment from MEGA X software. Phenotypic observations including grain number per panicle, root length, and days to emergence were taken from individual plants and seedlings grown in greenhouse or growth chamber conditions (16 h:8 h light:dark photoperiod, 27 °C). Photomicrographs of inflorescence tissues at five stages (from meristem to immature spikelets as described in Jiao et al., 2018) were collected and processed for scanning electron microscopy (SEM).

### 5.2. Transcriptome Profiling

Sample collection, processing, and transcriptomic profiling was conducted as described in Jiao et al. (2018). Three replicates (ten plants each) at each stage of panicle development were used for tissue collection. The ten plants for each replicate were processed as follows: at stage 1, whole panicles were harvested; at stage 3, differentiated floral organs on the tips of panicles were isolated; and at stages 4 and 5, the SS and PS tissues were isolated. For each replicate, the ten samples for each stage were pooled together. Tissues were immediately frozen in liquid nitrogen and stored at −80 °C prior to RNA extraction.

RNA was extracted using the TRIzol reagent, and then treated with DNase and purified using the RNeasy Mini Clean-up kit (Qiagen). Total RNA quality was examined on 1% agarose gels and RNA Nanochips on an Agilent 2100 Bioanalyzer (Agilent Technologies). Samples with RNA integrity number ≥ 7.0 were used for library preparation. Poly (A)^+^ selection was applied to RNA via oligo (dT) magnetic beads (Invitrogen 610-02) and eluted in 11 µl of water. RNA-seq library construction was carried out with the ScriptSeq™ v2 kit (Epicentre SSV21124). Final libraries were amplified with 13 PCR cycles. RNA-seq of three biological replicates was executed at the sequencing center of Cold Spring Harbor Laboratory on an Illumina HiSeq2500 instrument.

RNA-seq data from each sample was first aligned to the sorghum version 3.4 reference genome using STAR [57]. Quantification of gene expression levels in each biological replicate was performed using Cufflinks [58]. The correlation coefficient among the three biological replicates for each sample was evaluated by the Pearson test in the R statistical environment. After removal of two low-quality samples, the biological replicates were merged together for differential expression analysis using Cuffdiff [58]. Only genes with at least five reads supported in at least one sample were subjected to differential expression analysis. The cutoff for differential expression was an adjusted FDR of *p* < 0.05. Motif enrichment analysis was performed using the MEME SUITE [59]. GO term analysis was performed with either the agriGO [60] Singular Enrichment Analysis using the hypergeometric statistical test method with significance level set to 0.01, or the GO Enrichment Analysis using PANTHER version 11 with all default presets [61]. All raw FASTQ files have been deposited in the NCBI Sequence Read Archive (see Data Availability statement). Statistical analysis, including PCA biplots (factoextra package), heatmap generation (heatmap2), along with additional figure generation (ggplot2), was performed using RStudio v1.1.463 [62].

### 5.3. DAP-Seq Analysis

The full length *MSD1* coding sequence (CDS) was cloned into the pDEST15 Gateway vector, and the resultant plasmid was transformed into BL21 competent cells. GST-MSD1 protein was induced by growing cells in Terrific Broth at 28 °C while shaking at 220 rpm; isopropyl-beta-D-thiogalactoside (IPTG) (Goldbio: I2481C25) was added to a final concentration of 0.001 M. GST-MSD1 protein was purified by resuspending cells in 1x PBS + 10 mM phenylmethanesulfonyl fluoride (PMSF), and then sonicating at 4 °C to disrupt cell membranes and plasmid DNA. Soluble cell extracts were added to MagneGST beads (Promega) and incubated and washed as described in Bartlet et al. (2017) [63]. High-purity DNA was isolated from stage 4–5 developing BTx623 meristems and sheared on a Covaris S220 sonicator. Template DNA from three biological replicates was incubated with bead-bound GST-MSD1 protein or GST beads alone (negative control). The MSD1-bound DNA was then washed, eluted, and ligated with Illumina adaptor sequences and quality-controlled using Qubit and Bioanalyzer as described in Bartlet et al. (2017). Sequencing was performed using the mid-output from the Illumina NextSeq platform, multiplexing all six samples, yielding ~16–20 × 10^6^ reads per samples. Two separate sequencings runs were performed on experimental samples to increase detection power, with biological replicates undergoing 75-bp and 100-bp paired-end reads. The resultant FASTQ files were aligned and merged as follows: Trimmomatic [64] was used for FASTQ trimming, followed by Bowtie2 [53] alignment and MACS2 [65] peak calling (using the bead-only control for background subtraction), and finally the annotatePeaks program from the Homer [66] package was used to associate peaks with gene models from the version 3.4 *Sorghum bicolor* reference genome files housed by Gramene [67,68]. Sorghum GFF and GTF files were both used for annotatePeaks features functionality; however, the GTF file yielded more total gene-associated peaks than the GFF file. SAMtools was used for various file formatting and manipulation steps, including sorting and merging of the 75-bp and 100-bp paired-end read files. Motif enrichment analysis was performed using the MEME SUITE.

### 5.4. Phenotypic Rescue of msd2 with Exogenous Methyl-Jasmonate

Phenotypic rescue was performed exactly as described in Jiao et al. (2018). Briefly, BTx623 or *msd2* mutant seeds were germinated and grown at 16-h day cycles at 24 °C in a polyethylene greenhouse in Lubbock, TX. Beginning at leaf stage 7, 1 mL of either 0.05% Tween-20 (polyethylene glycol sorbitan monolaurate) in water (control) or 0.5 mM or 1.0 mM methyl-jasmonate in 0.05% Tween-20 was aspirated directly down the floral whorl. This treatment was repeated every 48 h until the majority of control plants reached the flag leaf stage. At that point, all experimental treatments were halted for that genotype. All plants were allowed to mature to the soft dough stage prior phenotypic rescue evaluation.

## 6. Data Availability

Sequencing data is available on the National Center for Biotechnology Information Sequence Read Archive (NCBI SRA: https://www.ncbi.nlm.nih.gov/sra). Accession codes for FASTQ files are as follows: DAP-seq, PRJNA550273; RNA-seq, SRP127741 [7] and PRJNA550261. DAP-seq BED files from MACS2 calls are available in Appendix A.

## Figures and Tables

**Figure 1 ijms-20-04951-f001:**
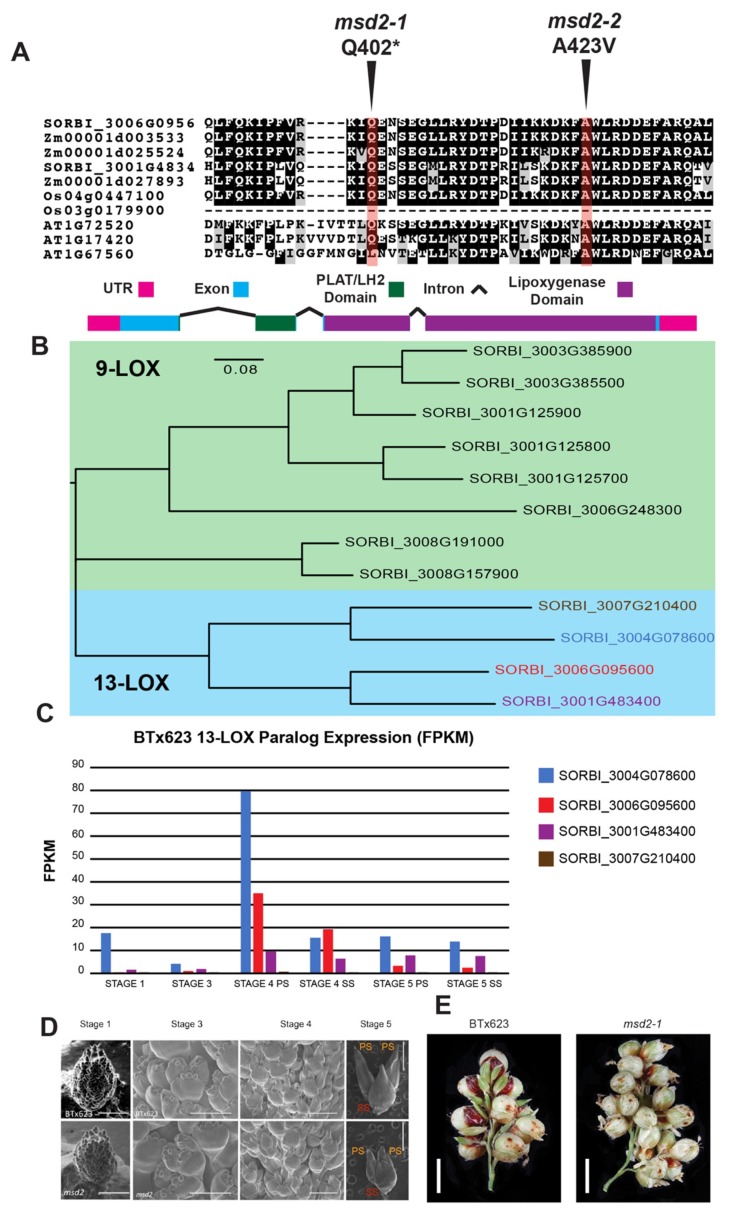
(**A**) Boxshade section of a MUSCLE alignment for MSD2-orthologous sorghum, maize, rice, and Arabdiopsis lipoxygenase (LOX) peptide sequences surrounding the ethyl methane sulfonate (EMS)-induced changes within MSD2. Arrows with red highlights indicate the positions of the *msd2-1* (GLN > premature stop) and *msd2-2* (Ala > Val) mutations. Below the alignment is a diagram of the *MSD2* gene; colored boxes indicate encoded domains of the gene product. The sizes of legend boxes are equivalent to 100 base pairs. (**B**) Phylogenetic tree of sorghum 9- and 13-LOX proteins (MSD2 highlighted in red). (**C**) RNA-seq FPKM expression data of the 13-LOX paralogs across developing panicle tissue stages (colors correspond to panel **B**). (**D**) SEM images of developing inflorescence meristems in WT and *msd2-1*. Scale bars are 1 mm in length for stages 1, 4, and 5, and 500 μm for stage 3. Sessile spikelets (SS) are indicated in red and pedicellate spikelets (PS) in orange. (**E**) A section of a secondary branch of late-dough filling panicles from WT and *msd2-1* lines. White scale bar indicates a length of 0.5 cm.

**Figure 2 ijms-20-04951-f002:**
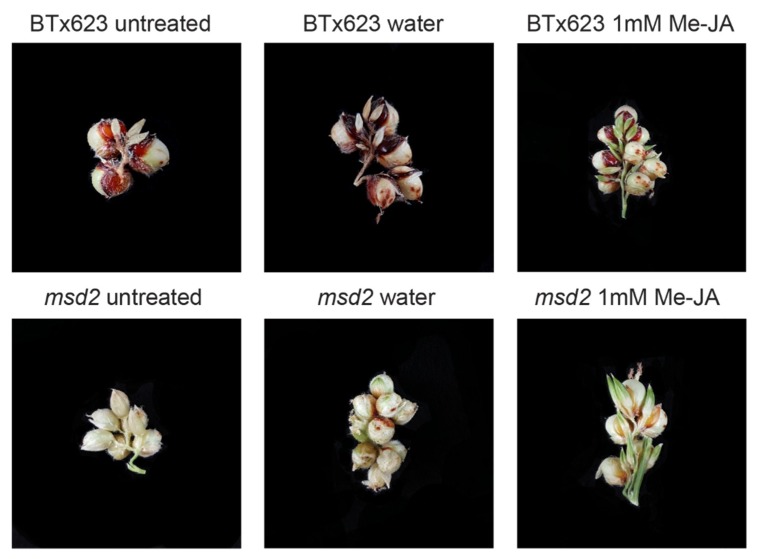
Phenotypic rescue of *msd2* plants with exogenous application of methyl-JA. Wild type (WT) and *msd2* lines were treated every 48 h with either water + 0.05% Tween-20 or 1 mM Me-JA + 0.05% Tween-20.

**Figure 3 ijms-20-04951-f003:**
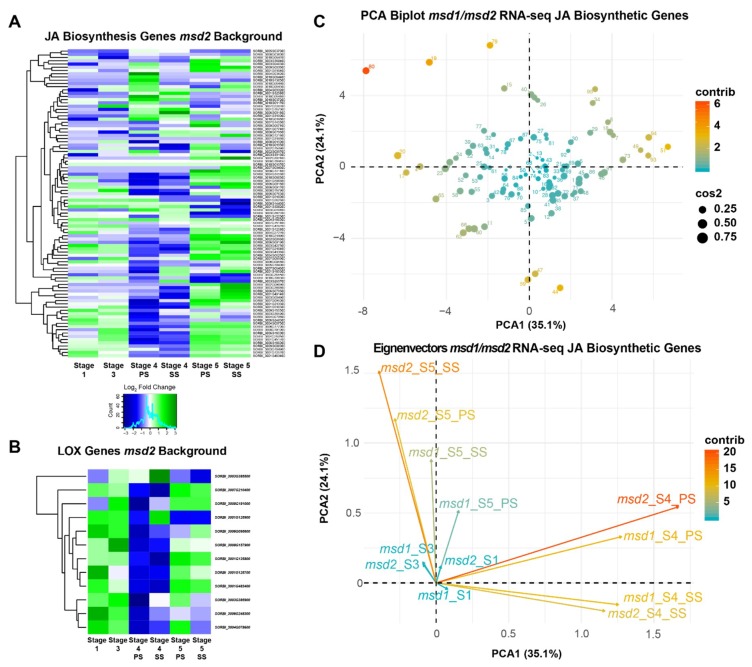
Transcriptomic profile showing the WT: *msd2* log_2_(fold change) of (**A**) Jasmonic Acid (JA) biosynthetic pathway genes and (**B**) only LOX paralogs (based on homology from *Arabidopsis* and maize orthologs) across various stages of meristem development. (**C**) Principal component analysis (PCA) biplot and (**D**) eigenvectors of meristem stages from *msd1* and *msd2* RNA-seq data for the JA biosynthesis genes. Plot points are colored and sized according to dimensional contribution and quality, respectively. Eigenvectors are colored according to dimensional contribution.

**Figure 4 ijms-20-04951-f004:**
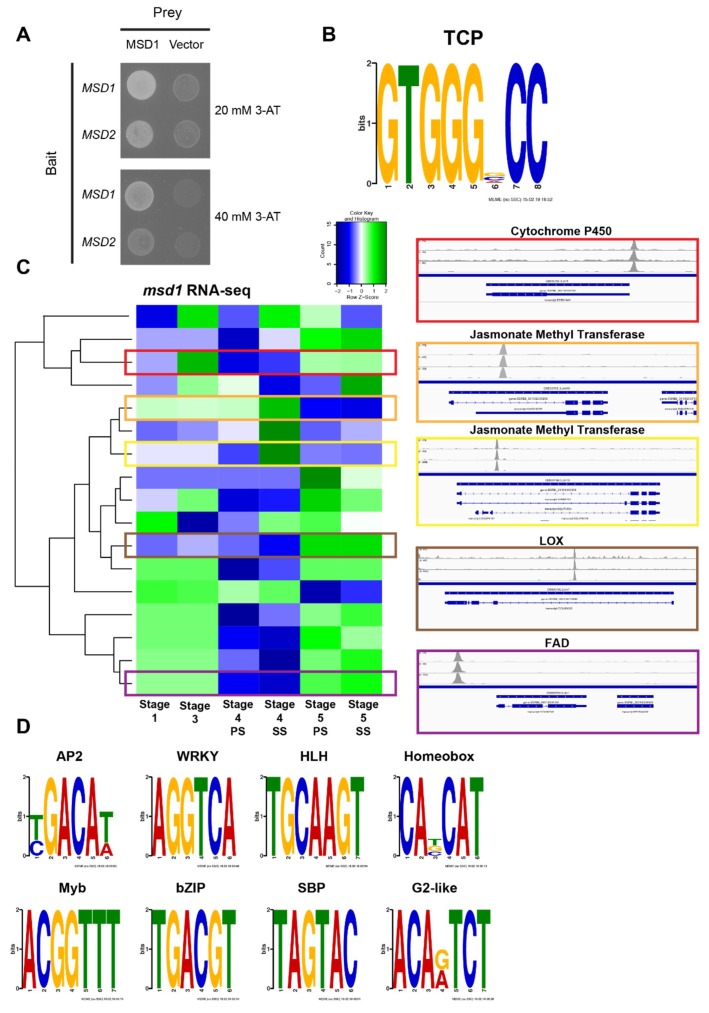
MSD1 as a regulator of developmental signaling genes. (**A**) Yeast 1-Hybrid of MSD1-mediated activation of gene expression by binding to the *MSD1* and *MSD2* upstream promoter regions. (**B**) Teosinte branched/Cycloidea/PCF (TCP) binding motif enriched in the MSD1 DNA Affinity Purification sequencing (DAP-seq) peaks that are localized within 2000 bp of an annotated gene transcriptional start site (TSS). (**C**) RNA-seq data from *msd1* showing downregulation across developing panicles stages in coordination with highlighted DAP-seq peaks localized near the transcriptional start sites (TSSs) of JA pathway genes. (**D**) Enriched DNA-binding motifs from all significant DAP-seq peak sequences, regardless of distance from an annotated gene TSS.

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
