# Peer review of "Fertility of Pedicellate Spikelets in Sorghum Is Controlled by a Jasmonic Acid Regulatory Module"

_ijms, 2019, doi:10.3390/ijms20194951_

Round 1

Reviewer 1 Report

The work of Gladman and colleagues investigates the mechanism that promotes the formation of fertile spikelets in sorghum. They characterize a mutant isolated from a previous EMS mutant collection that showed fertile spikelets in the place of the sterile pedicellate spikelets.

Using transcriptomics and molecular biology techniques (yeast on hybrid and DNA affinity purification) they uncover that the gene MSD2 whose mutation induce the phenotype is a lipoxygenase involved in jasmonic acid biosynthesis. Evidences suggest MSD1 regulates MSD2 directly binding the putative regulatory region.

The work concludes confirming that the jasmonate pathway is involved in spikelets fertility in sorghum and can be manipulate to increase the seed production up to 100% (although reducing the seed size).

The work is logically presented and easy to follow. The techniques used are of high quality and the results are of great impact for the scientific community. The experiments proposed represent a rigorous approach to the analysis of a commercially interesting plant.

I have few minor comments:

Pictures of spikes in Figure 1E, Figure2 are smaller than they should be. As they are very important for the study I don't see the reason for making then small including so much black background.

As the work includes also consideration of commercial value, it would be important to have estimation of seed size with a picture of the wt versus msd2.

It is of big interest if the seed size reduction concern only the seeds of the spikelets in place of pedicellate spikelets, or affects also the seeds of the other fertile spikelets.

Supplemental figure 1 A is very difficult to interpret as the samples in comparison are differently dissected and with different magnification. The same magnification and dissection would make easier to understand.

In conclusion I believe this work will be a useful source of study on sorghum development and its commercial exploitation.

Author Response

Dear Reviewer,

Thank you for your comments. We have addressed your edits as follows:

We modified the Figure 1E panels to increase the size of the panicle branches for emphases. I have also included a size bar as a comparison on seed size. Seed size is important. We had included a mass/100 seed count boxplot for msd2-1 and msd2-2 lines compared to WT BTx623 in Supplemenatry Figure 1. This, along with the size bar included in Figure 1E (as stated in point 1), should be sufficient for demonstrating average seed size. Seed size from the normally fertile sessile spikelet is indeed uniformly reduced in mutant compared to WT. This was meant to be conveyed in Line 118. We have added additional wording to clarify. We do not have flowering sorghum that is ready to dissect at this time; doing so would require another several weeks. We felt that performing a full dissection of the WT spikelet to show the absence would be more convincing alongside a more in-tact floral dissection of the msd mutant pedicellate spikelet.

Reviewer 2 Report

The authors of the manuscript "Fertility of pedicellate spikelets in sorghum is controlled by a jasmonic acid regulatory module" presented molecular mechanisms that which stand behind enhancing seed productivity of Sorgum bicolor. The manuscript is presented in high-quality, which is also the general evaluation rate of the scientific background. Some specific observation are listed below:

L34: please replace "JA" with "Jasmonic Acid" in the keywords. Abbreviations are not particularly welcome here.

L38-39: does this enzyme have additional effects on plant's general habitus or specific morphological characteristics? This might influence the overall productivity.

L47: please italicize "Sorgum bicolor". What is (1) in parenthesis standing for?

L60: replace "this" with "which" and finish the sentence after "fertilize". Start the next sentence with "They will..."

L87: please un-italicize the authority name "Moench".

L89: "PCF" insead of "PFC".

L92: expression "independent alleles" is quite questionable. Sure they derive from different genotypes, but I would ratehr say "different" instead of "independent".

L92-94: end of the sentence is lacking.

L118: delete "a". Plural for "gynoecium" is "gynoecia".

L125: "were" instead of "was"

L245: a "d" excessive in "Evaluation".

Author Response

Dear Reviewer,

Thank you for your comments. We have addressed your edits as follows:

L34: has been changed as suggested. L38-39: Not that we have quantitatively observed. The narrow temporal-spatial expression profile of MSD2 indicates that any effect would likely be manifested in the development of the panicle and probably not elsewhere. Qualitatively it appears that the msd2 mutant can result in a larger panicle head with increased branching, but we have not been able to quantify this in a satisfying way. We have not observed any other significantly different phenotypes from WT in field or greenhouse conditions. L47: Changed. The "(1)" indicated the first reference, but it was erroneously placed since it is also referenced at the end of the sentence. It has been removed. L60: Changed. L87: Changed. L89: Changed. L92: Changed to "different". We were intending to convey that the msd2 alleles (and other msd1 alleles) are independent EMS events as stated in line 104. L92-94: Added "followed by comparative variant calling analysis" to make it a complete experimental statement. L118: Changed. L125: Changed. L245: Changed.